# Advanced Hyperspectral Image Analysis: Superpixelwise Multiscale Adaptive T-HOSVD for 3D Feature Extraction

**DOI:** 10.3390/s24134072

**Published:** 2024-06-22

**Authors:** Qiansen Dai, Chencong Ma, Qizhong Zhang

**Affiliations:** 1School of Artificial Intelligence, Hangzhou Dianzi University, Hangzhou 310018, China; qiansendai@hdu.edu.cn; 2School of Computer Science and Technology, Hangzhou Dianzi University, Hangzhou 310018, China; machencong@hdu.edu.cn

**Keywords:** hyperspectral images (HSIs), image classification, tensor, low-rank representations, 3D feature extraction, multiscale, adaptive, SmaT-HOSVD

## Abstract

Hyperspectral images (HSIs) possess an inherent three-order structure, prompting increased interest in extracting 3D features. Tensor analysis and low-rank representations, notably truncated higher-order SVD (T-HOSVD), have gained prominence for this purpose. However, determining the optimal order and addressing sensitivity to changes in data distribution remain challenging. To tackle these issues, this paper introduces an unsupervised Superpixelwise Multiscale Adaptive T-HOSVD (SmaT-HOSVD) method. Leveraging superpixel segmentation, the algorithm identifies homogeneous regions, facilitating the extraction of local features to enhance spatial contextual information within the image. Subsequently, T-HOSVD is adaptively applied to the obtained superpixel blocks for feature extraction and fusion across different scales. SmaT-HOSVD harnesses superpixel blocks and low-rank representations to extract 3D features, effectively capturing both spectral and spatial information of HSIs. By integrating optimal-rank estimation and multiscale fusion strategies, it acquires more comprehensive low-rank information and mitigates sensitivity to data variations. Notably, when trained on subsets comprising 2%, 1%, and 1% of the Indian Pines, University of Pavia, and Salinas datasets, respectively, SmaT-HOSVD achieves impressive overall accuracies of 93.31%, 97.21%, and 99.25%, while maintaining excellent efficiency. Future research will explore SmaT-HOSVD’s applicability in deep-sea HSI classification and pursue additional avenues for advancing the field.

## 1. Introduction

Hyperspectral images (HSIs) are acquired through dedicated sensors such as AVIRIS, Hyperion, and CHRIS, capturing numerous spectral bands across Earth’s geography [1,2]. Distinct species exhibit spectral separability, reflecting unique spectral information characteristics [3]. Unlike conventional images, HSI extends beyond spatial features, offering detailed one-dimensional spectral profiles per pixel [4]. At the same time, hyperspectral imaging has a wide range of potential in various electromagnetic radiation frequency ranges, such as X-ray, ultraviolet, visible and near-infrared, and terahertz [5,6,7,8,9]. With applications spanning agriculture, geology, ecology, and mineralogy [10,11,12,13], HSI faces challenges in classification due to its high spectral resolution and noise, potentially leading to dimensionality disaster and reduced classification accuracy, especially with limited training samples [14,15]. Hence, efficient feature extraction is pivotal for HSI classification tasks.

Various feature extraction methods have emerged, from classical linear techniques like principal component analysis (PCA) and linear discriminant analysis (LDA) to nonlinear approaches such as neighborhood preserving embedding (NPE) and locality preserving projections (LPP) [16,17,18,19]. However, these methods often neglect spatial detail and concentrate solely on spectral information, limiting their effectiveness [20].

To address this, approaches integrating spatial and spectral information have gained prominence, categorized into segmentation-based, decision-based fusion, and feature fusion methods [21]. In segmentation-based methods, superpixelwise PCA (SuperPCA) and superpixelwise adaptive singular spectral analysis (SpaSSA) emphasize localized feature extraction with superpixel segmentation [22,23]. Decision-based fusion methods like robust matrix discriminative analysis (RMDA) combine multiple classification decisions for improved integration [19]. Yet, these methods often treat spectral and spatial information independently, overlooking their intrinsic connection within HSI [20].

Recognizing HSI’s cubic form encompassing spatial and spectral dimensions [21], feature extraction in 3D has emerged, notably through deep learning and tensor-based methods. Deep learning, exemplified by 3D convolutional neural networks (3DCNNs) and transformer-based networks like SpectralFormer and spectral-spatial transformer network (SSTN), has shown promise in extracting deeper features [24,25,26,27]. Despite the substantial progress made by the aforementioned methods, they share common challenges inherent in deep learning, such as the need to determine a large number of hyperparameters and the lack of model interpretability. The tensor-based approach exploits the low-rank structure of HSI, with algorithms such as Tucker decomposition and CANDECOMP/PARAFAC (CP) efficiently extracting 3D features [28,29]. Fu et al. [20] introduced tensor singular spectral analysis (TensorSSA) to extract global and low-rank features of HSI by constructing the global trajectory tensor of HSI. Nonetheless, the low-rank representation of the tensor is sensitive to initialization and parameter selection and is less universally applicable across all types of data, indicating its limitations.

One notable approach is truncated higher-order singular value decomposition (T-HOSVD), a form of Tucker decomposition that compresses tensors for denoising and approximate low-rank representation [30,31]. However, T-HOSVD faces challenges in handling diverse and complex HSI data, along with the difficulty of determining optimal ranks [32]. Therefore, it is necessary to explore an improved T-HOSVD method that can adaptively perform rank estimation while reducing data sensitivity and making the algorithm more robust for better 3D feature extraction of HSI.

To overcome these challenges, this paper proposes a novel method, Superpixelwise Multiscale Adaptive T-HOSVD (SmaT-HOSVD), which integrates local features and low-rank attributes of HSI for 3D feature extraction. Spectral and spatial features are extracted in 3D through superpixel and Multiscale Adaptive T-HOSVD to obtain more comprehensive low-rank information and reduce sensitivity to data, thereby enhancing classification tasks.

The primary contributions of SmaT-HOSVD include the following:(1)Introducing SmaT-HOSVD, a 3D feature extraction method utilizing superpixels and Multiscale Adaptive T-HOSVD, which fully utilizes spectral and spatial information and enhances the comprehensiveness of features;(2)Employing adaptive rank estimation based on the energy distribution of data, enhancing noise separation and inter-class distinction. In addition, a low-rank fusion strategy is utilized to generate multilevel features to enhance the comprehensiveness of the features and the robustness of the algorithm;(3)Demonstrating low time complexity and memory footprint, enhancing computational efficiency, and validating performance across multiple public datasets.

## 2. Related Works

We adopt italicized letters for scalars (e.g., *x* and *X*), bold letters for matrices (e.g., X), and calligraphic letters for tensors (e.g., X). ℝ denotes the real number fields. X∈ℝI1×⋯In×⋯×IN denotes a third-order tensor.

### 2.1. Notations and Definitions

**Definition** **1**(n-mode flattening matrix). *An n-mode flattening matrix is obtained by unfolding a tensor along its i_n_ dimension and then flattening it to a two-dimensional array. For a tensor* X∈ℝI1×⋯In×⋯×IN*, the resulting matrix is denoted as*
(1)Xn∈ℝIn×(I1×I2×⋯×In−1×In+1×⋯×IN)

**Definition** **2**(n-mode product [33]). *The n-mode product of a tensor*
X∈ℝI1×⋯In×⋯×IN
*with a matrix*
U∈ℝJ×In
*is denoted by*
γ=X×nU∈ℝI1×⋯×In−1×Ij×In+1×⋯×IN*, whose elements can be denoted by*
γi1i2⋯in−1jin+1⋯iN=∑in=1InXi1i2⋯in−1inin+1⋯iNUjin*. This operation can also be expressed through the matrix:*
(2)γ=X∗nU⇔Y(n)=UX(n)

**Definition** **3**(singular value decomposition (SVD) [34]). *For the matrix*
X∈ℝn1×n2*, the SVD of X is given by*
(3)X=U∑V*
*where* U *is an*
$n_{1}\times n_{1}$
*complex unitary matrix,*
∑
*is an*
$n_{1}\times n_{2}$
*rectangular diagonal matrix,* V *is an*
$n_{2}\times n_{2}$
*complex unitary matrix, and* V^*^
*is the conjugate transpose of* V.

### 2.2. T-HOSVD

The truncated higher-order singular value decomposition (T-HOSVD) [30] is a tensor decomposition technique that has been successfully applied to denoising [35], face recognition [36], and various other applications. Notably, T-HOSVD exhibits outstanding performance in HSI.

For HSI X∈ℝI1×I2×I3 with a multilinear rank denoted by rank-(R¯1,R¯2,R¯3), we can determine the optimal X¯ that approximates X by solving an optimization problem:(4)minX¯∈RI1×I2×I3∥X−X¯∥F2 s.t. rank-(R¯1,R¯2,R¯3)
where (R¯1,R¯2,R¯3) is the reduced multilinear rank, and the ∥•∥F is the Frobenius norm. To solve this optimization problem, we can employ a simple method, namely, T-HOSVD. The detailed steps of T-HOSVD are as follows.

Unfolding X obtains a matrix $X_{n}$ by n-mode, compute a rank-${\bar{R}}_{n}$ truncated SVD:(5)$$X_{n}=U_{n}\underset{n}{\sum }V_{n}^{T},\;n=1,2,3$$
and store the top ${\bar{R}}_{n}$ left singular vectors $U_{n}\in R^{I_{n}\times {\bar{R}}_{n}}$, then compute the core tensor $G=X\times_{1}U_{1}\times_{2}U_{2}\times_{3}U_{3}$ by the n-mode product. Finally, the approximation tensor can be reformulated as
(6)$$\bar{X}=G\times_{1}U_{1}\times_{2}U_{2}\times_{3}U_{3}$$

Tucker decomposition offers numerous advantages and finds applications across various fields [37]. De Lathauwer and Vandewalle [38] have investigated the application of the Tucker decomposition in the field of signal processing. Vasilescu and Terzopoulos [39] were pioneers in introducing the Tucker decomposition for applications in computer vision. As a specific instance of orthogonal Tucker decomposition, T-HOSVD serves as a generalization of matrix singular value decomposition. While T-HOSVD does not offer an optimal multilinear rank approximation [40], in practical applications, it tends to produce satisfactory solutions when optimal solutions with small errors exist. However, T-HOSVD encounters two significant challenges in the field of HSI processing. Firstly, HSI often contain diverse targets with multilayered and complex information, making T-HOSVD susceptible to variations in data distribution and reducing its robustness. Secondly, determining the optimal rank for an HSI remains a challenging problem. Typically, the optimal rank is computed by transforming rank estimation into an iterative optimization problem with constraints. However, this approach is time-consuming, and tuning certain parameters can be difficult. To enhance the solution and minimize errors, we incorporate adaptive low-rank estimation and multiscale fusion strategies in our approach. These strategies contribute to a more comprehensive and robust extraction of features from HSI, as detailed in Section 3.2.

## 3. Methods

The flowchart of the proposed SmaT-HOSVD method is depicted in Figure 1, comprising the following three parts: (1) Superpixel segmentation; (2) Superpixelwise Multiscale Adaptive T-HOSVD; (3) Classification. The details of the proposed SmaT-HOSVD are described as follows. 

### 3.1. Superpixel Segmentation

Superpixel segmentation involves partitioning pixels into several irregular pixel blocks known as superpixels [41]. The boundaries of these superpixels align relatively well with the actual boundaries of the image, facilitating subsequent task processing [42]. Superpixel algorithms can be broadly categorized into two types: graph-based superpixel algorithms and gradient descent-based superpixel algorithms. Classic algorithms include entropy rate superpixel (ERS) [43] and simple linear iterative clustering [41], which are widely used in HSI preprocessing.

ERS transforms the superpixel segmentation problem into solving the objective function with the largest entropy rate on the graph. In ERS, images are mapped to a graph, where each pixel is represented as a vertex, and edge weights represent pairwise similarities [42]. By solving the objective function with the largest entropy rate on the graph, k connected subgraphs are generated, where k is the number of superpixels. SLIC is akin to a clustering method, initializing the superpixel center and conducting local iterations based on the similarity of surrounding pixels, ultimately obtaining the superpixel. However, SLIC ignores global image properties and is insensitive to pixel content [42]. Considering these factors, we opted to use ERS to obtain superpixels.

In our method, principal component analysis (PCA) is applied to the HSI, and the first principal components (PCs), $I_{f}$, obtained are utilized for superpixel segmentation. This is because the first PCA contains the primary information of the HSI. By applying ERS to the $I_{f}$, the HSI can be segmented into superpixel blocks:(7)$$I_{f}=\overset{S}{\underset{k}{\cup }}X_{k},\;s.t.\;X_{K}\cap X_{g}=\emptyset ,\;(k\neq g)$$
where S denotes the number of superpixels, and $X_{k}$ is the kth superpixel.

### 3.2. Superpixelwise Multiscale Adaptive T-HOSVD

In reality, determining the rank T-HOSVD is not a straightforward task. Therefore, we prefer to employ a simpler method to adaptively determine the rank-$\left({\bar{R}}_{1},{\bar{R}}_{2},{\bar{R}}_{3}\right)$. During the SVD of a matrix, the larger singular values often contain more intrinsic information of the matrix. Based on this principle, we propose a simple strategy to automatically estimate the rank ${\bar{R}}_{n}$.

For HSI X∈ℝI1×I2×I3, unfolding X obtains a matrix Xn by n-mode. Let σi be the singular values corresponding to matrix Xn, arranged from large to small. Let C be the sum of singular values:(8)C=sum(σi), i=1,2⋯In

The rank ${\bar{R}}_{n}$ can be determined by
(9)$${\bar{R}}_{n}=\min(i),\;\mathrm{sum}(\sigma_{k}{)\;<\;T}_{n}C,\;k=1,2\cdots i$$
where T is a given threshold, $\mathrm{sum}(\sigma_{k}),\;k=1,2\cdots i$ is the sum of the first i singular values of $X_{n}$. Therefore, we can set the threshold-$\left(T_{1},T_{2},T_{3}\right)$ based on experience to adaptively determine the rank-$\left({\bar{R}}_{1},{\bar{R}}_{2},{\bar{R}}_{3}\right)$.

For a given HSI X, we can set different thresholds $T^{m},\;m=1,2,\cdots M$,where M is the number of different scales. For T-HOSVD of different scales, different structures and characteristics will be revealed [44]. Ranks of different sizes can reflect various levels of intrinsic information in HSI. Therefore, the rank of one scale often cannot entirely capture the information. It is necessary to integrate sufficiently complementary and relevant information. Multiscale fusion becomes an effective method.

Then, by superimposing the third dimension of $X^{m}$, the result of Multiscale T-HOSVD can be determined by
(10)$$\tilde{Y}=[X^{1},X^{2},\cdots ,X^{m}]\;\in R^{I_{1}\times I_{2}\times (I_{3}\times m)}$$

The $X^{m}\in R^{I_{1}\times I_{2}\times I_{3}},\;m=1,2,\cdots ,M$ represents the T-HOSVD representation at various scales. However, simply fusing T-HOSVD representations of different scales together will inevitably have negative effects. On one hand, the contributions of different scales are treated equally, while the importance of different scales is ignored; on the other hand, stacking the third dimension will cause a dimensionality disaster, which may significantly impact the classification results [45].

In order to better extract information at different scales and enable multiscale fusion to better display the intrinsic structure and characteristics of HSI, we can perform PCA on $\tilde{Y}$ to achieve dimensionality reduction:(11)$$y=\tilde{y}\times_{3}W^{T}$$
where W is the transformation matrix composed of the eigenvectors corresponding to the first n eigenvalues of the covariance matrix $Y_{3}Y_{3}^{T}$. Finally, we can obtain the result $Y\in R^{I_{1}\times I_{2}\times n}$ for the Multiscale Adaptive T-HOSVD.

By integrating HSI representations from adaptive T-HOSVD at multiple scales, complementary and relevant information can be effectively captured. Figure 2 illustrates the application of Multiscale Adaptive T-HOSVD to HSI. Among them, we use “Feature Image” color pictures to better represent the extracted features. Although HSI can capture a significant amount of useful information, its practical effectiveness is often limited due to the richness and complexity of the contained data. However, superpixel blocks derived from superpixel segmentation are highly homogeneous regions with similar structures and features. Therefore, applying Multiscale Adaptive T-HOSVD within these homogeneous regions tends to extract more effective features. We refer to this process as SmaT-HOSVD for superpixel blocks, as shown in Figure 3.

Since superpixels are typically irregular regions, preprocessing these irregularly shaped superpixels is necessary. For each superpixel, we need to identify the smallest bounding rectangle that encompasses the superpixel, which may include some adjacent pixels from other superpixels. All bands within this rectangular region are then extracted to form the 3D tensor required for Multiscale Adaptive T-HOSVD. This 3D tensor is processed using Multiscale Adaptive T-HOSVD, and the original superpixel is replaced with the resulting low-rank tensor.

This approach can effectively reduce intra-region differences while highlighting inter-class differences. Additionally, to the best of our knowledge, this is the first time an improved version of T-HOSVD has been applied to superpixel blocks. This method can adaptively estimate the optimal rank for different superpixel blocks and effectively utilize the spatial-spectral correlation of pixels in local images.

### 3.3. Classification

After feature extraction from the HSI to obtain the feature tensor, the next step is to feed it into the classifier for classification. Support vector machine (SVM) [46] is a classification algorithm whose basic idea is to map the data into a high-dimensional space and find a hyperplane in that space, maximizing the distance of all data points to the hyperplane. SVM has demonstrated good performance in the classification of remote sensing data [47]. Therefore, we decided to use SVM as our classifier.

In addition, the key of SVM is to choose the type of kernel function, mainly linear kernel, polynomial kernel, Gaussian radial basis function (RBF), and so on. Following the suggestion from [48], we decided to use RBF as our kernel function.

Algorithm 1 for the proposed method is shown below.
**Algorithm 1:** Superpixelwise Multiscale Adaptive T-HOSVD(SmaT-HOSVD)**Input:** Hyperspectral image X∈ℝI1×I2×I3, ground truth label L∈ℝI1×I2, number of superpixels S, reduction dimensionality n, threshold T^1^ and T^2.^
1:The first principal component X′ is obtained by applying PCA to X is obtained by applying PCA to X.2:Apply Equation (7) to get superpixels If3:While Xk in {If} do4:         Fill Xk to get the standard tensor Xk′5:         Apply Equations (8) and (9) to compute the optimal rank of Xk′6:         Apply Equations (5) and (6) for Xk′ to obtain the low-rank representations Y1and Y27:         Apply Equation (10) to get multiscale representation Y˜=[Y1,Y2]8:          Apply Equation (11) get results *y*9:         Update the superpixel Xk with the result Y with the result Y10:         k ← k+111:End while12:Updating X∈ℝI1×I2×n with superpixels If13:Training and testing of X and L using SVM**OUTPUT:** Classification results

## 4. Results

### 4.1. Datasets

In our experiments, we chose three publicly available hyperspectral datasets to evaluate the reliability of our model.

(1)Indian Pines: This dataset was acquired by the imaging spectrometer AVRIS over the Indian Pines test field in northwest Indiana. It has a spatial size of 145 × 145 pixels and encompasses a total of 224 bands of reflectance data. The spectral reflectance bands range from 400 nm to 2500 nm. Approximately two-thirds of the hyperspectral scene is dedicated to agricultural areas, while one-third is predominantly covered by forest or vegetation. To focus on relevant information, bands that do not capture water reflections were excluded, resulting in the utilization of the remaining 200 bands for the study.(2)Pavia University: Collected by the ROSIS sensor during a flight over Pavia, northern Italy, this dataset comprises 610 × 240 pixels in 115 consecutive bands within the wavelength range of 430 nm to 860 nm. Noise affects 12 of the bands, leading to the selection of the remaining 103 bands for the study.(3)Salinas: Captured by the imaging spectrometer AVRIS which is manufactured by NASA’s Jet Propulsion Laboratory (JPL), this dataset represents an image of the Salinas Valley in California, USA. The image size is 512 × 217, and it includes 224 bands. Bands that do not capture water reflections were excluded, resulting in the selection of the remaining 204 bands for the study. The dataset includes areas with vegetables, bare soil, and vineyards.

### 4.2. Experimental Setup

We employed SVM classifiers to validate the effectiveness of SmaT-HOSVD. Following the methodology of previous work [49], we utilized five evaluation metrics: producer accuracy (PA), overall accuracy (OA), average accuracy (AA), kappa coefficient (Kappa), and running time (s). To mitigate systematic errors and reduce random errors, all experiments were independently conducted ten times, and the average values were computed. Both training and test sets were randomly selected without any repetition in either. The training set represented 2% (Indian Pines), 1% (Pavia University), and 1% (Salinas) of the categories in each sample, respectively.

To validate the effectiveness of the proposed method, we compared it with seven state-of-the-art HSI feature-extraction methods. Considering that SmaT-HOSVD utilizes superpixel segmentation and tensor low-rank representation, we selected two methods based on superpixel segmentation, i.e., SuperPCA and SpaSSA, and one tensor-based method, TensorSSA. In addition, due to the wide application of deep learning, we also consider two deep learning methods, i.e., SpectralFormer and SSTN. Moreover, we also selected T-HOSVD and SVM to highlight the effectiveness of SmaT-HOSVD.

(1)A support vector machine (SVM) is employed as a classifier with a Gaussian radial basis function (RBF) serving as its kernel function. To determine the hyperparameters, we utilize five-fold cross-validation.(2)We utilize T-HOSVD for feature extraction and compare it with our proposed SmaT-HOSVD. The classifier employed is SVM, and the rank-$\left({\bar{R}}_{1},{\bar{R}}_{2},{\bar{R}}_{3}\right)$ is set to 60.(3)SpectralFormer [26] is a neural network based on transformer, utilizing a patch-wise version with a patch size of 1 × 3 and trained for 300 epochs.(4)SSTN is employed with a batch size of 32 and trained for 100 epochs, while keeping other hyperparameters consistent with the specifications in [27].(5)SuperPCA relies on dimensionality reduction through superpixel segmentation, employing the same parameters as specified in [22].(6)SpaSSA is grounded in feature extraction with superpixel segmentation, and the optimal parameters specified in [23] are utilized.(7)TensorSSA extracts spectral-spatial features through a 3D approach, and the optimal parameters specified in [20] are employed.(8)For our proposed method, the optimal parameters are detailed in IV.C. SVM is employed for classification.

The experiments were configured on MATLAB R2021a on a Windows 10 64-bit platform with an AMD Ryzen 7 5800H (3.2 GHz) CPU and 16 GB of memory which is manufactured by Advanced Micro Devices, Inc. (AMD). Deep learning implementation was carried out on the Ubuntu 20.04 platform using the PyTorch 1.10.2 framework with an NVIDIA GeForce RTX 3050Ti GPU.

### 4.3. Comparisons with State-of-the-Arts Models

Analyzing the classification results obtained using random selection and different training sets is crucial for understanding the impact of varying training data sizes on model performance. In this subsection, we conduct extensive experiments to compare the proposed method with state-of-the-art methods as detailed in Section 4.2.

(1)Performance with Different Training Percentages: Figure 4 displays the overall accuracy (OA) achieved by different methods using various training percentages for the three datasets. Specifically, the randomly selected training sample percentages vary as follows: 1%, 2%, 3%, 4%, and 5% for the Indian Pines (IP) dataset; 0.5%, 1%, 2%, 3%, and 5% for the Pavia University (PU) dataset; and 0.2%, 0.4%, 0.5%, 1%, and 2% for the Salinas (SD) dataset.

Figure 4 illustrates that the accuracy of all methods generally improves as the percentage of training samples increases. Notably, the proposed SmaT-HOSVD consistently outperforms seven comparative methods, particularly when dealing with small sample sizes (e.g., 1% IP, 0.5% PU, and 0.2% SD), achieving excellent performance. Comparatively, state-of-the-art deep learning method SSTN and feature extraction method SuperPCA yield suboptimal results across the three datasets, falling short of SmaT-HOSVD, highlighting the effectiveness and advancement of SmaT-HOSVD. Additionally, other methods exhibit varying performance across different datasets. For instance, TensorSSA performs well as the second-best method on the PU dataset after SSTN and SmaT-HOSVD, but its classification performance on the IP and SD datasets is mediocre. In contrast, SmaT-HOSVD consistently achieves the best results across all three datasets, underscoring the robustness and efficacy of the proposed method.

(2)Quantitative Evaluation: To thoroughly evaluate the effectiveness of the proposed method, we provide a detailed comparison of producer accuracy (PA) for each category in Table 1, Table 2 and Table 3, along with the three evaluation metrics OA, AA, and Kappa. The proposed SmaT-HOSVD consistently achieves the highest accuracy in most classes and outperforms other methods in terms of overall metrics. Taking the IP dataset in Table 1 as an example for analysis, firstly, in terms of OA, SmaT-HOSVD improves 27.83%, 27.78%, 33.08%, 4.24%, 1.81%, 14.35%, and 7.35% compared to SVM, T-HOSVD, SpectralFormer, SSTN, SuperPCA, SpaSSA, and TensorSSA, respectively. Also, on AA and Kappa, SmaT-HOSVD improved to varying degrees. It is worth noting that the PA of SuperPCA is not much different from SmaT-HOSVD, but the processing time of SuperPCA is shorter. This is because during the processing of SmaT-HOSVD, too many neighboring pixels are added to some small superpixel blocks, which introduces noise and results in a degradation of the effect. And compared to SuperPCA, the time complexity of SmaT-HOSVD is higher, but overall, from all three datasets, SmaT-HOSVD achieves better performance. On the PA of each class, SmaT-HOSVD s’ accuracy is basically at the top of the list, above 80% for 5 classes and above 90% for 11 classes. The superior performance of the SmaT-HOSVD method is also observed in Table 2 and Table 3 for the Pavia University and Salinas datasets, respectively.

When comparing methods, both SuperPCA and SpaSSA demonstrate higher classification accuracy compared to the classical SVM. This improvement can be attributed to their utilization of local features, with SuperPCA exhibiting particularly notable performance in this regard. Unfortunately, SuperPCA does not perform well on the PU dataset, possibly due to the limitations of PCA-based feature extraction, especially for small targets. The deep learning method SpectralFormer is less effective than SVM, primarily because of the limited training samples that prevent adequate model training. In contrast, SSTN achieves better results by leveraging spectral-spatial features more comprehensively. The tensor-based method TensorSSA effectively extracts low-rank information but does not effectively utilize important local features, resulting in moderate performance. In contrast, SmaT-HOSVD demonstrates the best performance across all datasets by effectively extracting both local and spectral-space features.

(3)Qualitative Evaluation: A diagram categorizing IP, PU, and SD data is illustrated in Figure 5, Figure 6 and Figure 7. The analysis primarily focuses on the IP dataset presented in Figure 5. Once again, the emphasis is on the Indian Pines (IP) dataset depicted in Figure 5. SVM and T-HOSVD exhibit significant categorization noise across all categories, characterized by an exceptionally high number of classification errors. The performance of SpectralFormer is unsatisfactory, showing high accuracy only in a few classes while remaining extremely noisy in most classes. This outcome is likely attributed to limited training samples and underfitting of the model. SSTN is significantly more effective, reducing speckle-like classification errors, but still exhibits misclassifications in certain areas. SuperPCA achieves a high accuracy rate primarily because superpixel blocks provide localized features and enhance intra-class similarity. However, some classes are still affected by contamination from speckles. Both SpaSSA and TensorSSA also exhibit significant classification noise.

Our extracted SmaT-HOSVD is able to efficiently extract both spatial and spectral features thanks to the superpixel block and low-rank representation. As shown in Figure 5j, Figure 6j and Figure 7j, good classification results are basically obtained for targets in large or regular categories. This is due to the fact that SmaT-HOSVD effectively utilizes the spatial information through the superpixel blocks and obtains the main spectral information through the low-rank representation, which enhances the intra-class consistency and highlights the inter-class differences. However, the classification results are poor in some class edges and irregular mini-classes, which is due to the fact that too many pixels from other classes are added to introduce noise during processing, making the extracted features not characterize the intra-class information well. Intuitive analysis shows that the SmaT-HOSVD method can effectively reduce the noise and improve the classification accuracy compared with other methods. 

(4)Analysis of Running Time: Table 1, Table 2 and Table 3 likewise give the running times for all the methods. From the table, it can be seen that T-HOSVD, SuperPCA, and SmaT-HOSVD are the fastest and basically belong to the same order of magnitude. The training time for SpectralFormer and SSTN is prolonged due to the model’s complexity and the high number of training iterations, even when utilizing GPUs for training. SpaSSA, on the other hand, exhibits a longer runtime because the feature extraction needs to be carried out individually for each band, resulting in significant time consumption. TensorSSA requires performing the decomposition of the trajectory tensor, which is a time-consuming process. In comparison to SuperPCA and T-HOSVD, SmaT-HOSVD requires more time due to the additional tasks involved. However, despite this, SmaT-HOSVD is relatively efficient and performs well in terms of computational efficiency compared to many other methods.

Overall, the proposed SmaT-HOSVD method strikes a balance between runtime efficiency and classification accuracy, making it a viable choice for HSI classification.

## 5. Discussion

### 5.1. Parameter Analysis

In the proposed SmaT-HOSVD method, certain parameters require manual configuration. These parameters include the number of superpixels (S) in ERS-based superpixel segmentation, the dimension (n) for PCA dimensionality reduction, and the threshold (T) in T-HOSVD. To evaluate the impact of these parameters and determine optimal values, we conducted parameter sensitivity experiments on SmaT-HOSVD.

(1)Number of superpixels (S): Through superpixel segmentation, we can obtain different homogeneous regions where similar pixels are more likely to fall into the same class [22]. The parameter S determines the level of segmentation for the superpixels. A larger S leads to finer segmentation with larger superpixels, whereas a smaller S results in coarser segmentation with smaller superpixels. The degree of superpixel segmentation significantly influences the extracted features. Due to variations in spatial resolution and dataset size, the optimal number of superpixels (*S*) differs for each dataset. Although ideal classification accuracy requires over-segmentation of the scene, due to the complexity of the distribution of land cover, specific analysis is still required [50]. As shown in Figure 8, the best results are taken at 30, 25, and 90 for S in our approach for Indian Pines, Pavia University, and Salinas datasets, respectively. For example, in the IP dataset, a number of superpixel S below 50 achieves good results, which is due to the fact that the homogeneous regions in the IP dataset are around 30. When S is too large, it makes the image over-segmented, resulting in the obtained superpixel blocks being fragmented and narrow, making the algorithm ineffective.(2)Reduction dimensionality (n): The dimensionality reduction n directly influences the amount of information retained in the data. A larger value of n generally retains more information but may also increase noise; conversely, a smaller n emphasizes the main components of information, potentially leading to the loss of important details. Because of the diversity in data, the optimal dimensionality reduction n varies for each dataset. The ideal reduction n effectively filters out noise while preserving essential information. As shown in Figure 9, in our approach, the Indian Pines, University of Pavia, and Salinas datasets work best when the reduced dimension n is set to 20, 20, and 15, respectively. Based on the downscaling results of different datasets, our algorithm achieves the best classification performance in the range of downscaling dimensions from about 15 to 25. This shows that for the obtained low-rank tensor, the dimension reduction is usually around 20 to leave the most dominant information and remove the noise to extract the better features, which is consistent with the conclusion in [45]. When the downscaling dimension is larger, the information retained is cumbersome and contains noise, making the effect decrease. At the same time, a downscaling dimension of around 20 on different datasets achieves better results, which further illustrates the stability of our algorithm.(3)Threshold T^1^ and T^2^: The choice of different thresholds (T^1^ and T^2^) significantly affects the extracted features. The larger threshold-$\left(T_{1},T_{2},T_{3}\right)$ results in a higher rank-$\left({\bar{R}}_{1},{\bar{R}}_{2},{\bar{R}}_{3}\right)$, preserving more common features of the HSI. The level of low-rank features extracted varies with different T. As T increases, more dimensions are retained in the tensor decomposition, resulting in the preservation of a larger amount of information. This leads to extracted features that tend to be richer; however, they may also contain some noise. On the contrary, as T decreases, the tensor decomposition retains fewer dimensions, and thus some details and features may be lost. Furthermore, to simplify parameter settings, both T_1_ and T_2_ are set to the same value, while T_3_ is taken relatively independently. And in order to obtain different levels of features, T^2^ is generally set higher than T^1^, with T^2^ as a larger threshold and T^1^ as a smaller one. In T-HOSVD, the sum of the top 10% or even 1% of the singular values accounted for more than 99% of the sum of all singular values. Based on this principle, we set T^1^ as [(0.8,0.8,0.8), (0.85,0.85,0.9), (0.8,0.8,0.9)], and T^2^ as [(0.8,0.8,0.95), (0.85,0.85,0.95), (0.9,0.9,0.95)]. The detailed results of parameter tuning are illustrated in Figure 10.

From Figure 10, it is evident that the larger threshold value of T^2^ has a more significant impact on the proposed method and is a key parameter, while T^1^ serves as an auxiliary parameter with less impact. For different datasets, the optimal parameters for T^1^ and T^2^ are different. In our method, T^1^ for the Indian Pines dataset is set to (0.8,0.8,0.9), and T^2^ is set to (0.85,0.85,0.95); for the Pavia University dataset, T^1^ is set to (0.8,0.8,0.8), and T^2^ is set to (0.8,0.8,0.95); and for the Salinas dataset, T^1^ is set as (0.8,0.8,0.8), and T^2^ is set as (0.9,0.9,0.95). Our proposed method exhibits robustness to thresholding, as evidenced by the minimal variation in OA on Indian Pines, which is only 0.018. This result underscores the idea that different thresholds can offer complementary structural information.

Determining the number of superpixels and the optimal rank is a critical and open question in the proposed method, which aims to segment HSI and perform low-rank representation. In this paper, we experimentally set the number of superpixels and determined the optimal rank using an adaptive estimation strategy to achieve the best performance. The segmentation scale of HSI varies in terms of the number of homogeneous regions across different datasets. Therefore, in our experiments, we determined the optimal number of superpixel blocks using a continuous adjustment method to adapt to the specific characteristics of each dataset. In fact, we can determine the rank based on the percentage of total energy, as implemented in (8) and (9). The process of determining the optimal rank can be likened to setting a threshold for a given HSI. Given that the initial singular values in singular value decomposition typically contain the majority of the energy, we can fine-tune the threshold to identify the optimal rank. Thus, the above method can determine the number of superpixels and the optimal rank for different datasets.

### 5.2. Ablation Study

To compare the effectiveness of the proposed SmaT-HOSVD against T-HOSVD, we incorporate two key enhancements: ERS for superpixel segmentation and a multiscale adaptive low-rank representation strategy for T-HOSVD. We aim to validate these operations across three datasets. In the pixel-processing stage, we evaluate two scenarios: using the raw HSI without superpixel segmentation versus using ERS for superpixel segmentation. For the low-rank representation stage, we compare the original T-HOSVD with T-HOSVD employing the multiscale adaptive strategy. Specifically, we set rank-$\left({\bar{R}}_{1},{\bar{R}}_{2},{\bar{R}}_{3}\right)$ to 60 for the original T-HOSVD. However, due to varying sizes and shapes of superpixel blocks, smaller blocks may not achieve the length dimension of 60. Therefore, for the low-rank representation of superpixel blocks, we set rank-$\left({\bar{R}}_{1},{\bar{R}}_{2},{\bar{R}}_{3}\right)$ to 10. In our experiments, we select 2% of samples from the IP, 1% of samples from the PU, and 1% of samples from the SD datasets for training.

As shown in Table 4, the combination of superpixel segmentation and multiscale adaptive T-HOSVD (SmaT-HOSVD) achieves the highest classification performance. Comparing the raw HSI with superpixel blocks after using T-HOSVD separately shows that the latter has a higher classification accuracy across all three datasets. This improvement can be attributed to the tendency of superpixel blocks to contain similar or identical class targets, enabling low-rank representation to mitigate noise interference and enhance intra-class similarity. This makes the extracted features more different from one class to another, highlighting the gaps between different classes and facilitating the subsequent classification task. Regarding the enhanced T-HOSVD with adaptive rank estimation and multiscale fusion strategy, the classification accuracies are significantly improved compared with the original T-HOSVD. This is due to the fact that adaptive rank estimation can dynamically adjust the optimal rank according to the target information distribution, while enhancing the classification task by fusing features from different scales to obtain a more comprehensive representation. Moreover, this adaptive rank estimation and multiscale fusion does not add too much computation and has high efficiency. 

As depicted in Table 4, Superpixelwise Multiscale Adaptive T-HOSVD outperforms traditional T-HOSVD across the three datasets, showcasing improved classification performance. The combination of superpixel segmentation and low-rank representation can better remove HSI noise and extract effective features [51,52]. The information contained in the superpixel blocks obtained by superpixel segmentation varies in shape and size, and a fixed rank setting may not accurately reflect the information distribution of each superpixel block. Therefore, the use of adaptive rank estimation and multiscale fusion is particularly favorable for the low-rank representation of superpixel blocks, which enables the extracted features to reflect the information of each homogeneous region, thus enhancing the effectiveness of the classification task. Overall, leveraging superpixel blocks and multiscale adaptive T-HOSVD (SmaT-HOSVD) estimation yields superior performance in HSI classification.

### 5.3. Analysis between SmaT-HOSVD and Classification

In this paper, we utilize a combination of SmaT-HOSVD and SVM classifiers to achieve ground object classification. SmaT-HOSVD is employed for 3D feature extraction to obtain spectral-spatial features, aiming to reduce inter-class noise and enhance intra-class consistency for improved classification effectiveness. SmaT-HOSVD operates as an unsupervised 3D feature extraction method on HSI superpixel blocks without the need for labeled training data. This method effectively enhances inter-class differences and promotes inter-class consistency, thereby enhancing classification performance. Whereas supervised feature-extraction methods have been shown to further improve classification performance [53,54], and the sample prior information included in SmaT-HOSVD is also worth exploring.

However, there is a potential risk associated with SmaT-HOSVD. It may inadvertently diminish differences between similar objects while increasing agreement between dissimilar objects, which can introduce noise and lead to misclassification. The superpixel blocks obtained from superpixel segmentation may contain a higher number of different objects, and the introduction of noise in the algorithm can unintentionally reduce the differences between similar objects while increasing the consistency between dissimilar objects, leading to errors in classification. This requires us to avoid the superpixel blocks containing too much noise by finding a more appropriate superpixel segmentation method or designing an adaptive superpixel number-estimation algorithm. Meanwhile, for narrow and irregular superpixel blocks, dividing them into regular tensors may contain too many dissimilar pixels, resulting in the extracted features containing noise and leading to poor classification. In the subsequent improvement, we will seek an adaptive division of superpixels strategy instead of simple division to avoid excessive noise leading to the degradation of results.

## 6. Conclusions

HSIs, with their inherent three-dimensional nature, possess both spectral and spatial characteristics. Traditional methods often focus on either spatial or spectral features, or they attempt a simple fusion of the two. Consequently, there is a growing trend towards the utilization of 3D or tensor methods in HSI classification. However, these approaches frequently concentrate on global features and may overlook homogeneous regions, which could provide more effective spatial and spectral features. To address these limitations, this paper introduces the SmaT-HOSVD method for 3D feature extraction in HSI.

The proposed SmaT-HOSVD method classifies HSI into homogeneous clusters through superpixel segmentation to obtain superpixel blocks. It then performs adaptive rank estimation and multiscale fusion on these superpixel blocks to acquire low-order representations for each homogeneous cluster, facilitating 3D feature extraction of qualitative areas. SmaT-HOSVD effectively utilizes local features and low-order representations to capture the low-rank intrinsic features of HSI, thereby reducing redundancy and noise. This enhances the similarity within similar classes and widens the gap between different classes. By extracting features from HSI in 3D, SmaT-HOSVD obtains spectral-spatial information that facilitates subsequent HSI classification tasks. SmaT-HOSVD surpasses several state-of-the-art methods on three public datasets with an impressive overall accuracy improvement of 12% to 20%. Moreover, even with very little training, SmaT-HOSVD can achieve good overall accuracy while maintaining low time complexity, which has great potential in practical applications.

In future work, we plan to explore enhanced methods for incorporating irregular superpixels into HOSVD. This involves avoiding the use of excessive noise pixels to fill irregular superpixels, ensuring appropriate handling of irregularities. In addition, in the deep-sea manganese nodule classification task, due to the spectral spatial changes of HSI and the instability of illumination in the deep sea [55], we will also study the application of SmaT-HOSVD in the deep-sea manganese nodule classification task to improve the classification accuracy.

## Figures and Tables

**Figure 1 sensors-24-04072-f001:**
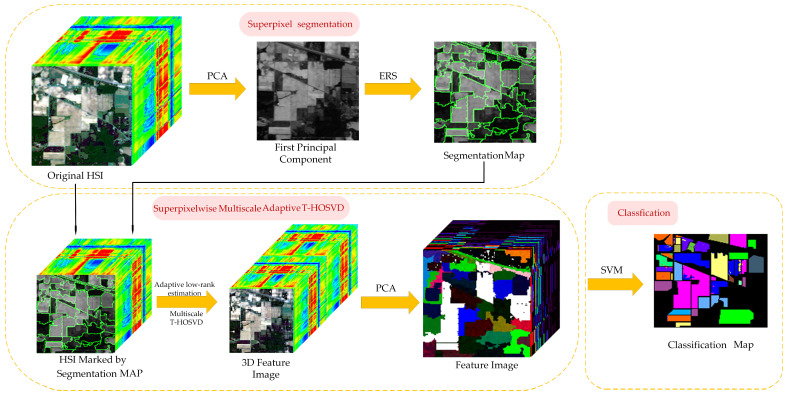
Schematic of the proposed SmaT-HOSVD-based HSI classification.

**Figure 2 sensors-24-04072-f002:**
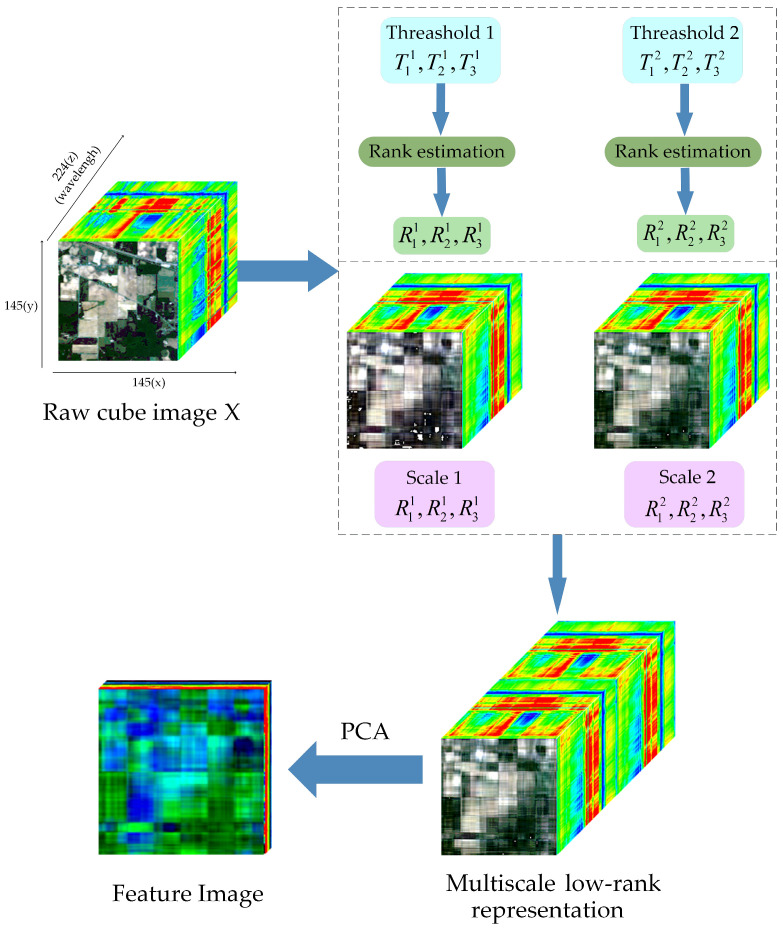
Schematic diagram of Multiscale Adaptive T-HOSVD usage on HSI Global.

**Figure 3 sensors-24-04072-f003:**
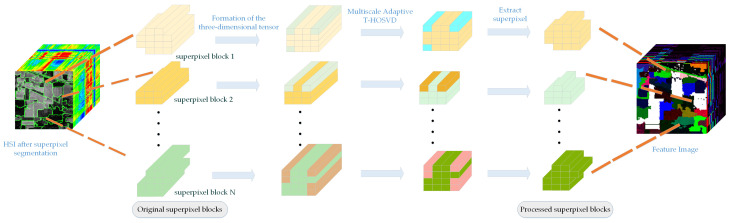
Schematic diagram of SmaT-HOSVD on HSI.

**Figure 4 sensors-24-04072-f004:**
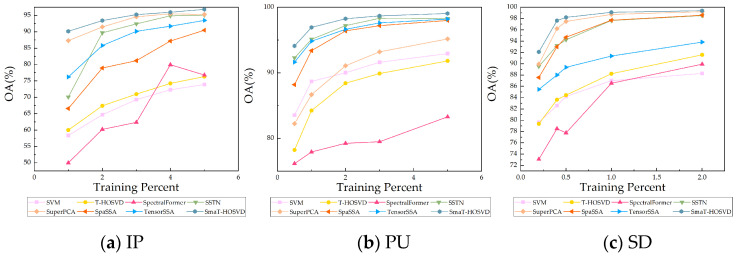
OA obtained by different methods with different training percentages over (**a**) IP, (**b**) PU, and (**c**) SD datasets.

**Figure 5 sensors-24-04072-f005:**
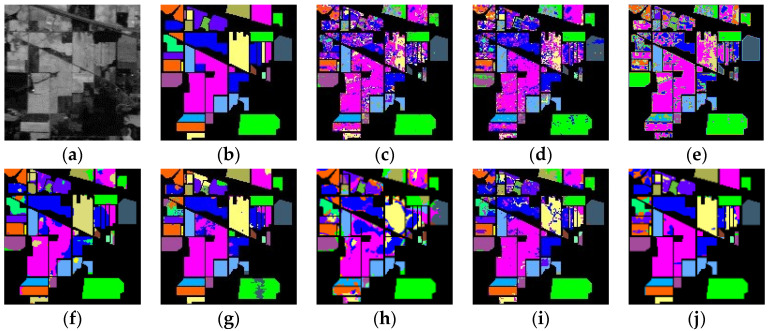
Classification results for the Indian Pines dataset (10% training samples): (**a**) Original; (**b**) Ground Truth; (**c**) SVM (66.09); (**d**) T-HOSVD (65.53); (**e**) SpectralFormer (60.23); (**f**) SSTN (89.79); (**g**) SuperPCA (90.9); (**h**) SpaSSA (75.14); (**i**) TensorSSA (85.22); (**j**) Ours (93.31).

**Figure 6 sensors-24-04072-f006:**
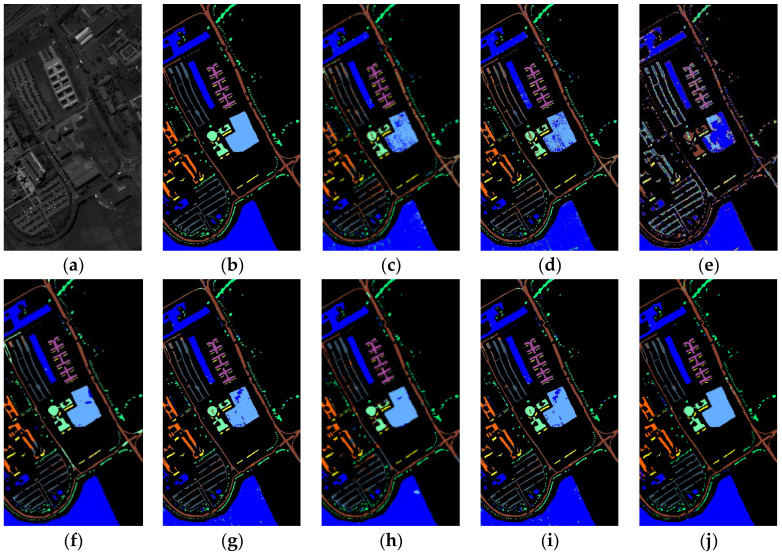
Classification results for the Pavia University dataset (1% training samples): (**a**) Origin; (**b**) Ground Truth; (**c**) SVM (88.81); (**d**) T-HOSVD (89.33); (**e**) SpectralFormer (77.94); (**f**) SSTN (95.12); (**g**) SuperPCA (85.34); (**h**) SpaSSA (93.61); (**i**) TensorSSA (94.78); (**j**) Ours (97.34).

**Figure 7 sensors-24-04072-f007:**
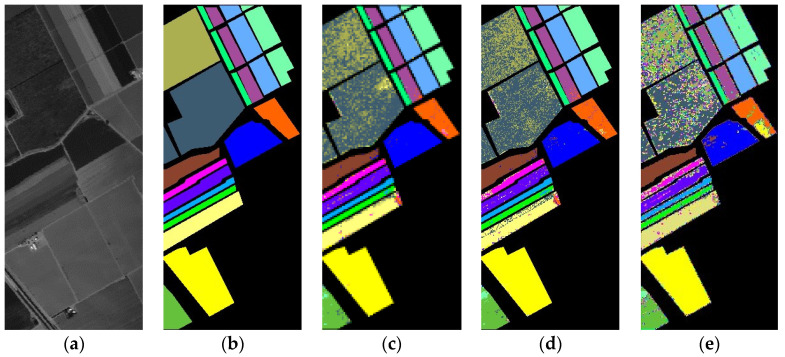
Classification results for the Salinas dataset (1% training samples): (**a**) Original; (**b**) Ground Truth; (**c**) SVM (86.47); (**d**) T-HOSVD (85.54); (**e**) SpectralFormer (86.55); (**f**) SSTN (97.66); (**g**) SuperPCA (98.89); (**h**) SpaSSA (97.06); (**i**) TensorSSA (90.00); (**j**) Ours (99.24).

**Figure 8 sensors-24-04072-f008:**
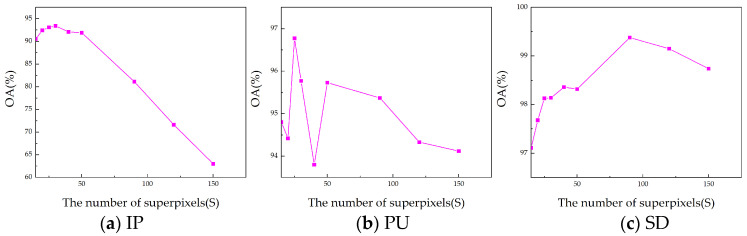
The influence of parameter S on overall accuracy (OA) concerning (**a**) IP with 2% training, (**b**) PU with 1% training, and (**c**) SD with 1% training.

**Figure 9 sensors-24-04072-f009:**
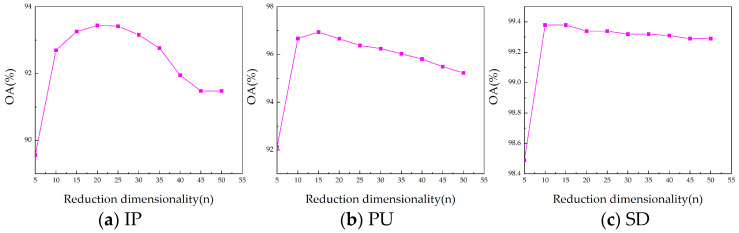
The influence of reduction dimensionality n on overall accuracy (OA) concerning (**a**) IP with 2% training, (**b**) PU with 1% training, and (**c**) SD with 1% training.

**Figure 10 sensors-24-04072-f010:**
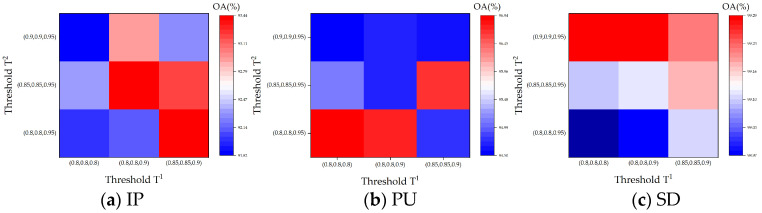
The influence of parameter T^1^ and T^2^ on overall accuracy (OA) concerning (**a**) IP with 2% training, (**b**) PU with 1% training, and (**c**) SD with 1% training.

**Table 1 sensors-24-04072-t001:** Classification results obtained by different methods for the IP dataset (2% training percentage).

Class	Samples	Svm	T-hosvd	Spectralformer	Sstn	Superpca	Spassa	Tensorssa	Ours
**1**	46	5.33	8.88	0.0	82.61	**100.0**	38.67	89.11	80.00
** 2 **	1428	57.13	68.40	51.75	**93.98**	90.26	75.16	81.49	87.59
**3**	830	45.87	38.62	38.50	**91.09**	85.00	74.61	83.92	84.76
**4**	237	18.32	21.55	33.62	**90.29**	61.25	81.16	67.98	80.52
**5**	483	69.28	66.38	20.08	82.19	91.86	76.53	85.12	**92.01**
**6**	730	84.77	81.53	89.37	93.01	96.24	90.04	94.80	**99.89**
**7**	28	27.04	59.25	0.0	**100.0**	96.3	74.44	95.19	95.93
**8**	478	94.92	87.39	99.78	**100.0**	96.94	90.56	93.27	**100.0**
**9**	20	24.74	31.57	0.0	**100.0**	95.26	28.95	99.47	84.74
**10**	972	56.17	44.43	38.44	89.81	88.05	74.14	81.82	**90.50**
**11**	2455	74.82	77.96	74.51	80.93	**95.67**	79.78	87.55	95.40
** 12 **	593	32.75	27.53	16.70	87.52	81.51	56.99	69.50	**90.65**
**13**	205	91.45	90.50	66.5	**100.0**	99.5	79.65	95.5	99.55
**14**	1265	89.99	93.94	97.34	98.18	97.27	92.71	97.37	**99.98**
**15**	386	25.90	93.94	15.61	79.02	94.37	69.79	74.89	**95.27**
**16**	93	49.78	39.56	83.52	49.46	54.29	86.81	85.38	**96.70**
**OA**		65.48	65.53	60.23	89.07	91.50	78.96	85.96	**93.31**
**AA**		53.02	53.76	45.36	88.63	88.99	73.12	86.40	**92.09**
**Kappa**		60.18	60.07	53.77	87.59	90.30	76.01	84.01	**92.37**
**Time (s)**		3.017	1.645	224.73	96	**1.11**	109.43	32.55	2.667

Bold represents the best result, and the color represents the different categories.

**Table 2 sensors-24-04072-t002:** Classification results obtained by different methods for the PU dataset (1% training percentage).

Class	Samples	Svm	T-hosvd	Spectralformer	Sstn	Superpca	Spassa	Tensorssa	Ours
**1**	6631	85.6	88.84	88.01	89.47	80.72	91.28	94.26	**96.47**
** 2 **	18,649	96.0	96.41	98.12	**99.89**	96.47	99.58	98.82	99.32
**3**	2099	67.8	78.72	3.90	84.75	86.57	77.82	84.22	**95.2**
**4**	3064	84.8	87.10	74.81	**89.55**	55.94	85.00	89.19	87.28
**5**	1345	98.9	98.87	98.87	**99.70**	98.14	99.17	99.19	98.43
**6**	5029	75.5	78.88	21.72	95.64	93.61	97.26	91.15	**98.91**
**7**	1330	78.7	75.30	6.16	**99.84**	79.29	75.59	92.03	96.43
**8**	3682	80.1	75.47	91.85	86.97	73.42	85.85	88.52	**96.83**
**9**	947	99.8	99.67	**100.0**	97.52	44.21	72.54	97.01	89.47
**OA**		88.1	89.33	77.94	95.12	86.68	93.39	94.27	**97.21**
**AA**		85.3	86.59	64.83	93.70	78.71	87.14	92.94	**95.37**
**Kappa**		84.1	85.76	69.23	93.51	82.15	91.21	92.97	**96.30**
**Time (s)**		12.2	4.731	513.2	118	**3.27**	859.0	34.33	11.298

Bold represents the best result, and the color represents the different categories.

**Table 3 sensors-24-04072-t003:** Classification results obtained by different methods for the SA dataset (1% training percentage).

Class	Samples	Svm	T-hosvd	Spectralformer	Sstn	Superpca	Spassa	Tensorssa	Ours
**1**	2009	97.41	98.34	98.44	99.69	100.0	98.40	98.27	**100.0**
** 2 **	3726	99.12	97.93	97.23	96.10	99.80	98.73	99.69	**99.82**
**3**	1976	89.79	92.99	78.43	99.54	97.88	99.19	97.75	**100.0**
**4**	1394	98.49	95.28	**99.20**	99.06	93.11	92.86	98.39	98.99
**5**	2678	96.67	96.15	97.10	69.34	96.51	96.04	97.24	**98.69**
**6**	3959	99.11	99.43	99.62	97.41	99.94	99.87	99.29	**99.95**
**7**	3579	**99.38**	99.37	98.08	75.28	99.27	97.23	99.33	99.29
**8**	11271	73.52	79.47	78.64	99.18	99.54	94.48	83.04	**99.59**
**9**	6203	98.81	98.97	98.37	**100.0**	99.67	99.57	99.13	99.92
**10**	3278	84.99	79.35	88.41	**100.0**	96.90	94.44	90.55	97.28
**11**	1068	87.08	83.34	71.14	97.46	90.65	98.15	90.25	93.29
** 12 **	1927	95.58	95.43	93.18	**100.0**	97.83	99.40	99.14	99.88
**13**	916	97.05	98.34	97.02	**100.0**	98.38	97.73	97.74	98.54
**14**	1070	89.49	87.15	91.50	**97.46**	97.07	96.02	91.47	95.94
**15**	7268	58.69	49.15	60.67	97.88	99.43	93.64	74.39	**99.49**
**16**	1807	89.16	89.14	86.69	**100.0**	99.24	95.93	94.00	99.15
**OA**		86.02	85.54	86.55	97.66	98.74	96.55	91.28	**99.25**
**AA**		90.90	89.99	89.61	94.69	97.83	96.98	94.35	**98.74**
**Kappa**		84.43	83.84	85.01	97.40	98.59	96.16	90.30	**99.17**
**Time (s)**		19.834	9.214	554.5	123	**3.602**	821.94	37.188	14.681

Bold represents the best result, and the color represents the different categories.

**Table 4 sensors-24-04072-t004:** OAs (%) of different combinations of pixel processing and low-rank representation on three datasets.

Pixel Processing		Low-Rank Representation Stage		Accuracy (OA)		
Raw HSI	Superpixel Blocks (ERS)	T-HOSVD	Multiscale Adaptive T-HOSVD	IP (2%)	PU (1%)	SD (1%)
✓		✓		62.9	87.58	84.76
✓			✓	66.39	89.44	86.64
	✓	✓		90.78	91.61	98.90
	✓		✓	93.31	97.21	99.25

## Data Availability

The hyperspectral remote-sensing images from the Indian Pines dataset, Pavia University dataset, and Salinas dataset were obtained from https://www.ehu.eus/ccwintco/index.php/Hyperspectral_Remote_Sensing_Scenes#Indian_Pines, https://www.ehu.eus/ccwintco/index.php/Hyperspectral_Remote_Sensing_Scenes#Pavia_University_scene, https://www.ehu.eus/ccwintco/index.php/Hyperspectral_Remote_Sensing_Scenes#Salinas_scene (accessed on 21 May 2024).

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
