# Peer review of "Advanced Hyperspectral Image Analysis: Superpixelwise Multiscale Adaptive T-HOSVD for 3D Feature Extraction"

_sensors, 2024, doi:10.3390/s24134072_

Round 1
Reviewer 1 Report
Comments and Suggestions for Authors
This paper introduces an unsupervised Superpixelwise Multiscale Adaptive T-HOSVD (SmaT-HOSVD) method, an improved version of the Truncated Higher-Order SVD (T-HOSVD), designed to determine the optimal order and address sensitivity to changes in data distribution. Several methods are compared, and multiple public datasets are used to validate performance. ​The organization of the manuscript is clear, and the figures are well-crafted. However, there are still several issues in the paper, thus I suggest a major revision to this paper.
1. In section 1: The use of abbreviations in the manuscript is problematic. Authors should provide the full names for abbreviations such as SSA, ERS, and SLIC upon their first appearance in the text.
2. In section 3: The novelty of the proposed method is not clearly highlighted, and the language used to describe the methodology requires further improvement.
3. In section 4.2: The rationale for selecting the seven methods for comparison with SmaT-HOSVD is not sufficiently explained. The authors should provide a detailed justification for choosing these specific methods.
4. In section 4.3: The data analysis is unclear. For example, in Table â… , the Producer Accuracy (PA) of superPCA is similar to that of SmaT-HOSVD, yet the processing time is shorter for superPCA. The authors should analyze and discuss the reasons for this observation.
5. In section 4: There is no analysis of the results and advantages of SmaT-HOSVD in terms of spectral and spatial features. This aspect should be addressed.
6. In section 5: The discussion lacks references to other studies. The authors should compare and discuss their results in the context of existing literature.
Comments on the Quality of English LanguageThe organization of the manuscript is clear, and the figures are well-crafted. However, certain language aspects could be improved. For example, the method description requires further refinement to enhance clarity and precision.
Reviewer 2 Report
Comments and Suggestions for Authors
1. First of all, the manuscript entitled "... for 3D Feature Extraction", but I did not find impressive visual results of 3D feature extraction. I believe they should be represented as a 3D image or something like that. Please add any images of the extracted 3D features.
2. In the introduction, near lines 35-36, I recommend mentioning also the wide potential of hyperspectral imaging in various frequency ranges of electromagnetic radiation; namely, in:
- X-ray [https://doi.org/10.1007/s10909-020-02456-9];
- ultraviolet [https://doi.org/10.3390/s21217332];
- visible and near-infrared [https://doi.org/10.1364/JOT.90.000706];
- near-infrared [https://doi.org/10.3390/
app13095226] and
- "terahertz" [https://doi.org/10.1364/OE.27.018456] frequency ranges.
3. lines 139-140: these steps "1) Superpixel segmentation; 2) Superpixelwise Multiscale Adaptive T-HOSVD; 3) Classification" are not indicated in Fig. 1. Please correct this.
4. The text should be carefully checked. For example, all abbreviations should be decrypted at the first mention. There are also some abbreviations, which are unnecessarily repeated (e.g. HSI). Please, check all the text again.
5. All figures should be improved. Let's consider Fig. 2 in detail: Please explain the red and black colors, as well as the other colors (e.g. blue and cyan). Moreover, the idea of adaptivity is not readable from Fig. 2. It should be immediately clear, which variable is adapting, according to which image' features it is adapted, and so on.
6. Datasets: Please provide the links to the datasets you have used:
https://paperswithcode.com/dataset/indian-pines and so on...
7. Consider the potential applications and implications of the SmaT-HOSVD in real-world tasks. And describe clearly the practical significance of the results, it provides.
Comments on the Quality of English LanguageMinor editing of English language required. The manuscript has problems, like this:
"However, these methods often neglect spatial information and concentrate solely on spectral information, limiting their effectiveness" - this sentence repeats the word "information". Select a synonym or rephrase.
Round 2
Reviewer 2 Report
Comments and Suggestions for Authors
The manuscript became much better after revision.
The remaining points are minor:
1. New added text on Fig. 1 ("1) Superpixel segmentation; 2) Superpixelwise Multiscale Adaptive T-HOSVD; 3) Classification") has low contrast since it is written in yellow color on brown background. I recommend to improve the contrast.
2. Remaining comment on Point 5: "All figures should be improved. ...Please explain the red and .... the other colors (e.g. blue and cyan)." Figure 2 was described as an example. And it was seriously improved. But some problems still take place:
- The color maps are not explained in the figures or figures' capture.
- 3D axes in hyperspectral datacube are not designated. I recommend adding the coordinate axes (x,y, "lambda"), with certain values on each axe: Probably, it will be pixels for the x, and y axes, and units on the wavelength-axis.
- In Figure 1, the colors in colormap over the wavelength axis in the datacube overlap with feature colors on the Feature Image (e.g. green, blue, and red), and with colors on the Classification Map. Do these colors mean the same? Probably, no, because the Feature Image and Classification Map also contain other colors (e.g. orange, brown).
- Explain yellow (bright and saturated ones), cyan, and bright green colors in Figure 3. Is the length of the 1D arrays over wavelength scale always the same, or for some data, it can be varied?
- Do the colors in the Classification Map coincide with the colors of the various classes, shown in Tables 1-3?
- Please denote the variable that corresponds to the enumeration of the spatial-spectral regions from the set indicated by the series of four black dots in Figure 3. I also recommend describing the elements of the figure more explicitly.
